# Silver Nanostars-Coated Surfaces with Potent Biocidal Properties

**DOI:** 10.3390/ijerph17217891

**Published:** 2020-10-28

**Authors:** Lucinda J. Bessa, Miguel Peixoto de Almeida, Peter Eaton, Eulália Pereira, Paula Gameiro

**Affiliations:** LAQV/REQUIMTE, Departamento de Química e Bioquímica, Faculdade de Ciências, Universidade do Porto, 4169-007 Porto, Portugal; mpda@fc.up.pt (M.P.d.A.); peter.eaton@fc.up.pt (P.E.); efpereir@fc.up.pt (E.P.); agsantos@fc.up.pt (P.G.)

**Keywords:** antimicrobial surfaces, biocidal properties, *Pseudomonas aeruginosa*, *Staphylococcus aureus*, star-shaped silver nanoparticles, silver nanoparticles

## Abstract

Bacterial proliferation on certain surfaces is of concern as it tends to lead to infectious health problems. Nanotechnology is offering new options for engineering antimicrobial surfaces. Herein, the antibiofilm and biocidal properties of star-shaped silver nanoparticles (AgNSs) in suspension and as coating surfaces were studied. AgNSs and spherical silver nanoparticles (AgNPs) (used for comparison purposes) were synthesized using reported methods. Glass disks (9 mm diameter) were covered with AgNSs using deposition by centrifugation. Minimum inhibitory concentrations (MICs) of AgNSs and AgNPs were determined against several reference strains and multidrug-resistant isolates and their antibiofilm activity was assessed against preformed biofilms of *Pseudomonas aeruginosa* and *Staphylococcus aureus* by both Live/Dead staining and atomic force microscopy (AFM). The antimicrobial properties of AgNSs-coated surfaces were evaluated by the “touch test” method on agar, and also Live/Dead staining and AFM. The MIC values of the AgNSs were 2–4 times lower than those of the AgNPs. Biofilms treated with AgNSs at a concentration equal to the MIC were not significantly affected, although they exhibited more dead cells than the non-treated biofilms. The biocidal activity of AgNSs-coated surfaces was attested, since no growth on agar nor viable cells were observed after contact of the inoculated bacteria with the coated surface for 6 and 24 h. Thus, AgNSs show greater potential as a surface coating with biocidal effects than used as suspension for antimicrobial purposes.

## 1. Introduction

Microorganism proliferation on certain surfaces in the hospital setting and in some industries is of concern as it tends to lead to infectious health problems. The problem is particularly worsened when those microorganisms form biofilms on such surfaces.

The complex microbial community within a biofilm is markedly more resistant to antibiotics and sanitizers than planktonic bacteria, leading to persistent survival that is a challenge to overcome [1]. Preventing the biofilm formation should be accomplished by regular cleaning and disinfection; however, due to several factors, some microorganisms are able to persist and occasionally develop a biofilm. Many studies have been made in order to develop new and more effective strategies to hamper biofilm formation, such as inhibiting the attachment of microbes by selecting surface materials that do not promote attachment, by modifying the surfaces’ physiochemical properties or by incorporating the antimicrobial products in the surface materials themselves [2].

Nanotechnology is offering new options for achieving better and more effective disinfectants and biocides [3,4], and for engineering antimicrobial surfaces [5,6].

Silver nanoparticles are well-known broad-spectrum antimicrobial agents [7,8]. Their strong antibacterial activity is believed to be related to simultaneous mechanisms acting synergistically in distinct bacterial targets. According to Tang and Zheng [8], silver nanoparticles affect cell membrane, membrane proteins, and DNA by (1) direct binding, (2) release of silver ions and (3) generation of reactive oxidative species. Chemical and physical properties such as stability, size, shape, and surface chemistry are known to have influence on the extent of antibacterial activity [9].

Silver nanoparticles are composed of silver atoms (Ag^0^); however, these atoms can be converted into silver cations (Ag^+^), which are then released from the surface of the particle to the medium (typically aqueous). This conversion depends on the redox conditions of the medium, and the velocity of this phenomenon can go from an instantaneous reaction to an ion-by-ion, layer-by-layer, release. Though silver nanoparticles with a spherical shape are the most common, various other shapes have been engineered, including stars, rods, platelets and cubes [10]. Star-shaped silver nanoparticles, or silver nanostars (AgNSs), which were used in this study and prepared as previously reported [11], display a core from which several arms are extended, and the length of those arms can be modified by controlling the concentration of reagents and the reduction time.

Herein, AgNSs suspensions were prepared and used for (i) assessing antibacterial and antibiofilm activities and (ii) further coating 9 mm diameter glass coverslips. Therefore, the initial assays consisted in determining (i) minimum inhibitory concentrations (MICs) of suspensions of AgNSs and of spherical silver nanoparticles, or silver nanospheres—AgNPs, (used for comparison purposes) against several reference strains and multidrug-resistant isolates; and (ii) their antibiofilm activity against preformed biofilms of *Pseudomonas aeruginosa* ATCC 27853 and *Staphylococcus aureus* ATCC 25923 by both fluorescence microscopy after Live/Dead staining and atomic force microscopy (AFM). Subsequently, the ultimate goal was to evaluate the antibacterial/biocidal action of AgNS-coated surfaces by means of several methods, namely the “touch test” method and subsequent colony forming unit (CFU) counting [12], fluorescence microscopy and AFM.

By covering a substrate with AgNSs, silver is readily available to be released under the form of Ag^+^ and thus able to exert its antibacterial effect. Compared to silver mirrors, coating with silver nanoparticles drastically increases the surface area of the substrate. For AgNSs, this is even more noticeable given the more evident nanoroughness of the surface. Additionally, star-shaped nanoparticles have a higher surface area to volume ratio when compared to spherical nanoparticles.

The environmental risk and toxicity associated with silver nanoparticles is a concern regarding the use of these nanomaterials. Toxicity is determined by the accumulation and bioavailability of the silver nanoparticle in each organism, and not just by the total concentration in the environment. Thus, biological factors can also influence toxicity as well as the size and shape and of the nanoparticles and the chemical properties of the medium, typically aqueous [13]. Since the extension of the consequences of these nanoparticles in aquatic ecosystems is quite unpredictable, the silver nanoparticles-based materials should be included in the implemented silver-containing waste treatment pathways, commonly used in healthcare facilities for discarding the X-ray films, silver nitrate patches used in wound care, etc.

## 2. Materials and Methods

### 2.1. Synthesis and Characterization of Silver Nanoparticles

Silver nanospheres (AgNPs) were synthesized as described elsewhere [14], without modifications.

Silver nanostars (AgNSs) were synthesized as described elsewhere [11], with some modifications. Briefly, 45 mL of a 1 mmol·dm^−3^ silver nitrate solution were added dropwise to a mixture of 2.5 mL of a 60 mmol·dm^−3^ hydroxylamine solution and 2.5 mL of a 50 mmol·dm^−3^ sodium hydroxide solution under stirring. After 2 min, 500 µL of a 1.5% wt. sodium citrate solution were added to the mixture. After 3 h under stirring, the grey-colored suspension was centrifuged for 15 min, using a 1600× *g* force. The supernatant was discarded, and the pellet resuspended in ultrapure water (up to 10% of the initial volume). The final suspension was stored in the dark and at 4 °C upon further use.

Nanoparticle tracking analysis (NTA) was performed using a Malvern Panaltyical NanoSight NS300 instrument, equipped with a 642 nm laser module (Malvern, UK). Data collection and analysis were performed using NTA 3.2 software, to obtain the hydrodynamic diameter distributions and the concentration of nanoparticles in the sample. The samples were diluted down to the 1–3 pmol·dm^−3^ range, using ultrapure water, and an aliquot of the diluted suspension was injected into the equipment’s flow cell. The focus and camera level were adjusted to obtain the best possible view of the particles, following the guidelines provided by the manufacturer. Five videos of 60 s each were captured, at different portions of the aliquot, enabling the measurement of a larger number of different particles. The analysis settings, in particular the detection threshold, were set depending on the scattered light intensity that was observed in the captured videos. The measurements were performed at 25 °C. Each video was analyzed independently, and the results were merged into a final particle size distribution chart.

### 2.2. Quantification of Silver in Nanoparticles

Inductively Coupled Plasma-Atomic Emission Spectrometry (ICP-AES) analysis was used for the quantification of silver (in mass) present at the suspensions of silver nanoparticles. For that, an aliquot of both suspensions of nanoparticles (AgNSs and AgNPs) was oxidized using nitric acid and hydrogen peroxide. The solutions resulting from the digestion of the nanoparticles were then analyzed by ICP.

### 2.3. Surface Coating with Silver Nanostars

A 1.5% agar solution was prepared, and 8 mL were added to 15 mL centrifuge tubes. After gelification, this solid agar worked as support for the glass surfaces (microscopy cover glass disks, 9 mm diameter, 0.13–0.16 mm thickness). Then, 500 µL of a 0.1 nmol·dm^−3^ AgNSs suspension and 1 mL of methanol were added into the tubes, over the glass surfaces. The prepared tubes were then centrifuged using a swinging-bucket rotor at a relative centrifuge force of 2500× *g*. The glass surfaces, uniformly coated, were gently removed from the tubes, washed with ultrapure water, left to dry at room temperature and finally stored upon further use. The AgNSs-coated surfaces were analyzed by scanning electron microscopy (SEM) using a FEI Quanta 400 FEG microscope.

### 2.4. Determination of Minimum Inhibitory Concentrations (MICs) of AgNSs and AgNPs

The MIC of AgNSs was determined in Mueller–Hinton broth cation adjusted (MHBII) as recommended by Clinical & Laboratory Standards Institute (CLSI) guidelines [15,16] and also in tryptic soy broth (TSB), which is the medium further used in biofilm assays. AgNSs were tested in the range of 0.5–0.08 nmol·dm^−3^ against four reference strains, *Escherichia coli* ATCC 25922, *Pseudomonas aeruginosa* ATCC 27853, *Staphylococcus aureus* ATCC 25923 and *E. faecalis* ATCC 29212, following the methodology of CLSI [15]. After 20 h of incubation at 37 °C, the MIC was read by the naked eye, being the lowest concentration that completely inhibited the bacterial growth (absence of turbidity). Afterwards, using TSB medium, the MIC assay was performed for AgNSs as well as for AgNPs (both from stocks at 1 nmol·dm^−3^ and assayed in the range of 0.5–0.08 nmol·dm^−3^) against the same strains but also against several multidrug-resistant clinical isolates of *P. aeruginosa* (PA006 and Pa4) and of methicillin-resistant *S. aureus*—MRSA (Sa1 and SA007). A silver nitrate solution (AgNO_3_), from a stock of 50 mmol·dm^−3^, was simultaneously assayed in the range of 10–0.01 mmol·dm^−3^.

### 2.5. Effect of AgNSs and AgNPs Suspensions on 24-h Biofilms—Microscopic Analyses

#### 2.5.1. Microscopy of Fluorescence—Live/Dead Staining

Biofilms of *P. aeruginosa* ATCC 27853 and of *S. aureus* ATCC 25923 were grown in TSB medium for 24 h on μ-Dish (35 mm, high)ibidi Polymer Coverslips (ibidi GmbH, Planegg-Martinsried, Germany), from starting inocula of 1 × 10^6^ CFU/mL. After 24 h, biofilms of each bacterial species were rinsed with Phosphate Buffer (PB) 1 mmol·dm^−3^ and filled with TSB + PB (control) or treated with (1) AgNSs 0.125 nmol·dm^−3^ (from stock of 1 nmol·dm^−3^; dilution of 8× in TSB) or (2) AgNPs 0.125 nmol·dm^−3^ (from stock 1 nmol·dm^−3^; dilution of 8× in TSB). Biofilms were again incubated for another 24 h and then rinsed 1× with PB 1 mmol·dm^−3^, stained with the Live/Dead solution (Invitrogen LIVE/DEAD BacLight Bacterial Viability Kit, Thermo Fisher Scientific, Portugal) for 30 min in the dark and rinsed again with PB 1 mmol·dm^−3^. Biofilms were then visualized by fluorescence microscopy (widefield microscope, Zeiss Axiovert 200 M), at a magnification of 630×.

#### 2.5.2. Atomic Force Microscopy

Biofilms of *P. aeruginosa* ATCC 27853 and of *S. aureus* ATCC 25923 were equally prepared and treated as mentioned above, with the exception that they were formed on Thermanox circular (15 mm diameter) plastic coverslips (Thermo Scientific, Rochester, NY, USA) placed in 35 mm diameter polystyrene plates. These were rinsed with PBS, dried, and examined in air. The incubated coverslips were examined using a TT2-AFM from AFMWorkshop. The scanning was carried out in vibrating (tapping) mode, using ACT cantilevers from AppNano and a 50 × 50 × 17 µm scanner. At least six areas per sample were imaged, to obtain representative results, and typical images are shown here.

### 2.6. Antimicrobial Testing of AgNSs-Coated Surfaces

Fresh grown colonies of *P. aeruginosa* ATCC 27853 and *S. aureus* ATCC 25923 were used to prepare bacterial suspensions with an OD_600_ = 0.1 (approximately 10^8^ colony-forming unit [CFU]/mL). Quantities of 50 μL of each suspension were pipetted on top of round glass coverslips (9 mm diameter), coated and non-coated with AgNSs. Three replicates of each condition and for each strain were performed. The antimicrobial properties of nanostar-coated surfaces were evaluated by the (i) “touch test” method and subsequent CFU counting after 6 and 24 h [12], (ii) microscopy of fluorescence after Live/Dead staining after 6 and 24 h and (iii) AFM after 24 h.

Briefly, through the “touch test”, after the incubation time (6 or 24 h) at 37 °C in a humidified incubator, bacteria from the AgNSs-coated surfaces were stamped on the top of agar plates for 30 s and then the surfaces were removed. The agar plates were then incubated at 37 °C for 20 h and the number of CFUs, when applicable, was determined.

For the microscopic analysis of the viability of bacteria previously inoculated on the top the AgNSs-coated and non-coated surfaces, after 6 or 24 h, the surfaces were stained with 50 µL of the Live/Dead solution for 30 min in the dark, rinsed with 100 µL PB 1 mmol·dm^−3^, and then visualized under the fluorescence microscope.

For AFM, the inoculated AgNSs-coated and non-coated surfaces were analyzed after 24 h of incubation and scanned as described above.

## 3. Results and Discussion

### 3.1. Characteristics of Silver Nanoparticles (AgNSs and AgNPs)

AgNSs were synthesized by the chemical reduction of Ag^+^ by neutral hydroxylamine, followed by a capping-reduction by citrate [11]. AgNSs used in this work had a hydrodynamic diameter mean of 186 nm, showing a high dispersity (Appendix A), with a multiplicity of sizes and a number of arms with different lengths, as observed by SEM (Appendix A).

AgNPs were synthesized by following a seeded-growth approach via the reduction of Ag^+^ by the combination of two reducing agents: citrate and tannic acid [14]. The synthesized AgNPs had a hydrodynamic diameter mode of 47 nm and showed lower dispersity in comparison to AgNSs (Appendix A).

The concentration of silver, as measured by ICP-AES, was 2400 μg/mL in AgNSs (1 nmol·dm^−3^) and 31 μg/mL in AgNPs (1 nmol·dm^−3^). Thus, the [Ag] is approximately 78 times higher in the AgNSs suspension than in the AgNPs one.

### 3.2. Antibacterial Activity of AgNSs Suspensions

The MIC values of AgNSs retrieved in the two media assayed were not similar. Lower MIC values were obtained when TSB medium was used (Table 1). Moreover, MIC values in MHBII varied according to the bacterial species, while in TSB the same MIC value was obtained for all four bacterial strains tested. It is known that certain components (mostly proteins) of complex media can interact with the nanoparticles, altering their physicochemical properties, and thus impacting the biological response [17,18].

Thereafter, TSB medium was selected to also determine the MIC of both AgNSs and AgNPs against multidrug-resistant clinical isolates (Table 2). The main reason to choose TSB was because it is the medium usually used in the biofilm formation assays, which we aimed to conduct next.

MICs were determined based on the concentration (nmol·dm^−3^) of the nanoparticles. The silver concentration on AgNSs and AgNPs suspensions was determined by ICP later on. As shown in Table 2, the MIC values of the AgNSs (nmol_AgNSs_·dm^−3^) were 2–4 times lower than those of the AgNPs (nmol_AgNPs_·dm^−3^). These results were expected given the fact that AgNSs suspension had a higher concentration of silver than AgNPs suspension, as revealed by ICP and demonstrated in Table 2. In fact, the Ag concentration at MIC value was 20–40 times higher for AgNSs than AgNPs and that is a rather huge difference, which can be explained by the following facts: (i) we do not know how much of the silver present in AgNSs is being released in the form of silver ions, which are likely to be mostly responsible for the antibacterial effect; and (ii) the AgNSs suspension tends to deposit after a while in the bottom of the wells of the microplate use in the MIC determination assay, which means less contact between the AgNSs and the bacterial cells which are suspended in the well and also less silver ions may be released from aggregated nanostars.

At the MICs values, the silver concentration was lower for AgNO_3_ compared to AgNSs and AgNPs. Moreover, the ratio of MICs values for [Ag^0^] (in AgNSs or in AgNPs) and [Ag^+^] (in AgNO_3_) was not constant for the several bacteria tested. This suggests that the antibacterial effect may not be explained considering exclusively the amount of silver ions on the liquid media, as others have also concluded [19]. However, such a measurement was not performed and, thus, the real silver ion release from the AgNSs is still an open question.

### 3.3. Mild Effect of AgNSs on Bacterial Biofilms

In order to evaluate the effect of both AgNSs and AgNPs on preformed biofilms of *P. aeruginosa* and *S. aureus*, microscopic analyses were carried out by fluorescence microscopy after Live/Dead staining and by AFM. Images of control and treated biofilms are shown in Figure 1. Biofilms were treated with AgNSs at 0.125 nmol·dm^−3^ (300 μgAg/mL), corresponding to the MIC for both strains, and equivalent biofilms were also treated with AgNPs at 0.125 nmol·dm^−3^ (3.9 μgAg/mL), which, because the exact MIC (MIC > 0.5) could not be determined, represents a sub-MIC value. Live/Dead images showed few dead cells (red cells) within the AgNSs-treated biofilms of both strains studied in comparison to the non-treated biofilms. AgNPs-treated biofilms were still totally viable, which was, in fact, as expected since the concentration used corresponded to a sub-MIC. By AFM, damaged bacterial cells could be observed within the AgNSs-treated biofilms of *S. aureus*, while in the AgNSs-treated biofilms of *P. aeruginosa*, bacterial cells were more difficult to be observed within the biofilm, thus hampering the analysis of their traits, as shown in Figure 1. Overall, no significant differences were observed between AgNSs-treated biofilms and the respective non-treated biofilms (controls), for both strains assayed. It is known that biofilms are more difficult to eradicate than planktonic bacteria; therefore, it is not surprising to see a mild effect of the AgNSs (at a concentration equal to the MIC) on 24-h preformed biofilms. Nonetheless, since the MIC of AgNSs was rather high in terms of μgAg/mL in comparison to the AgNPs, and also because of the visible dark color of the AgNSs suspensions, which would hamper their application at higher concentrations for a proper microscopic analysis, we did not test the effect at higher concentrations. Moreover, higher concentrations of AgNSs would provide the release of silver at concentrations that might harbor environmental risk and even toxicity [13].

Thus, then we focused on the main goal of this study, which was to use AgNSs suspensions to prepare coated surfaces and assess their biocidal activity, whose results are reported next.

### 3.4. Biocidal Effect of AgNSs-Coated Surfaces

Glass surfaces were uniformly coated with AgNSs, as seen by the naked eye (Appendix A). As shown by SEM analysis (Appendix A), some small gaps in the film were observed. We used image analysis to determine that an estimated 98.5% (S.D. 0.62 %) was covered by the film of stars. The coating was mechanically stable towards rinsing (3–4 times of rinsing did not cause alterations); however, it could be damaged in contact with other surfaces and detached by scraping.

The biocidal potential of these films was assessed by the “touch test” method [12]. The coated surface allowed complete killing of both *P. aeruginosa* and *S. aureus* after 6 h of contact of the inoculum with the coated surface (Figure 2), with zero CFU recovered. However, from non-coated surfaces, bacteria (both *P. aeruginosa* and *S. aureus*) could regrow with uncountable visible CFUs (Figure 2). After 24 h, the biocidal effect of AgNSs-coated surfaces was maintained, as also illustrated in Figure 2. Some “stains” on agar plates as a result of the “touch test” performed for AgNSs coated samples can be observed, but they do not represent bacterial growth; they occurred as a result of the AgNSs coatings detached from the samples.

The effect of the antibacterial effect of the AgNSs-coated surfaces was also evaluated microscopically after Live/Dead staining (Figure 3) and by AFM (Figure 4). The bacteria were inoculated on the surfaces (coated and non-coated) and incubated for 6 and 24 h. The inoculated surfaces were then, (i) after 6 and 24 h, stained with the Live/Dead solution and observed by microscopy of fluorescence or, (ii) after 24 h, directly observed under an atomic force microscope.

Bacteria showing green fluorescence have intact membranes (are viable), while red bacteria have damaged membranes (are dead). As we can observe in Figure 3, *P. aeruginosa* stained majorly red in non-coated and totally red in AgNSs-coated surfaces after 6 h and entirely red after 24 h, meaning that they were not viable anymore. However, a crucial difference was observable, which was the number of bacteria present. In non-coated surfaces the number was visibly higher, which means that the initial inoculum deposited on the surface was able to grow and proliferate; on the contrary, on AgNSs-coated surfaces few red bacteria are pinpointed, meaning that those initially inoculated on the surface were quickly killed and unable to multiply. The unexpected high number of red bacteria after 6 h on non-coated surfaces means that they were recently damaged; this can probably be explained because after those times of incubation, the inoculum (50 µL) placed on the surface dried out over time, leaving *P. aeruginosa* bacterial cells more fragile, being easily impaired throughout the process of Live/Dead staining. Such a hypothesis is supported by the fact that *P. aeruginosa* colonies were grown when the inoculated non-coated surface was stamped on the agar through the “touch test”, where we could see the regrowth (Figure 2). Moreover, it has been reported that Gram-positive bacteria may survive on dry surfaces for more extended periods than Gram-negative bacteria [20,21]. In the case of *S. aureus*, bacteria placed on top of the non-coated surfaces were totally viable after 6 h and had clearly proliferated. After 24 h, some were already dead. Nonetheless, on AgNSs-coated surfaces, same bacteria still were viable after 6 h. However, in this same condition, through the “touch test”, no regrowth on agar was seen. Most likely, those that still viable (green) entered a viable but non culturable state (VBNC) [22]. In the VBNC state, bacteria are unable to grow on routine culture medium but are actually alive and can resuscitate into a culturable state if favorable conditions are restored. Bacteria can enter the VBNC state as an adaptation to stress, be it drying, high temperatures, irradiation, antibiotics or disinfectants, but the VBNC can also be a state preceding cell death [23,24]. That might have been the case of *S. aureus* inoculated on AgNSs-coated surfaces, since after 24 h all bacteria stained red and were also not recovered on agar.

AFM images (Figure 4) showed that after 24 h, in non-coated surfaces, some *P. aeruginosa* cells were still intact, but others were already visibly altered, with an irregular morphology (Figure 4A). These results corroborated those from the “touch test”, thus, *P. aeruginosa* were still viable on non-coated surfaces after 24 h, while those *P. aeruginosa* cells placed on AgNSs-coated surfaces were totally fragmented and damaged (Figure 4B). *S. aureus* seemed quite intact after 24 h on non-coated surfaces (Figure 4C), and differences in the morphology could not also be seen on *S. aureus* cells placed on AgNSs-coated surfaces. By comparison with the data in Figure 3, we can conclude that *S. aureus* after 24 h on AgNSs surfaces, despite being dead, still showed a quite intact morphology. Little or no modification in *S. aureus* morphology, when in the presence of antibacterial agents at killing concentrations, is frequently observed [25,26]. Being a Gram-positive bacterium, its thicker peptidoglycan layer of the cell wall provides structural strength, not allowing gross morphological modifications to be seen.

## 4. Conclusions

The bactericidal effects of silver ions are well known. In this work, we used silver nanoparticles (nanospheres AgNPs and nanostars-AgNSs) as a “reservoir” of Ag^+^, under the form of silver atoms (Ag^0^) that comprise the nanoparticles and, when oxidized, can leave the particle as Ag^+^. Though AgNSs suspensions had a higher concentration of silver than AgNPs, the MIC values of AgNSs were 20 to 40 times higher than those of the AgNPs. This suggests that probably fewer silver ions are being released from AgNSs than from AgNPs into the medium and reaching the bacteria.

At the MIC, AgNSs had no effect on preformed biofilms of both *P. aeruginosa* and *S. aureus*. At that concentration, AgNSs target the bacterial cell and not the biofilm.

AgNSs suspensions were also used to coat glass disks. Each resulting coated surface presented a homogeneous distribution of AgNSs across the surface, as observed by SEM. The bactericidal effect of the AgNSs-coated surfaces was observable soon after 6 h of contact with the inoculated bacteria, *P. aeruginosa* and *S. aureus*. The bacteria deposited on the top of the AgNSs were not able to proliferate and were mostly killed after 6 h and totally dead after 24 h.

Therefore, the AgNSs described in this work showed greater potential in being used to coat surfaces, which become strongly biocidal, than in being used as suspension for antimicrobial purposes. AgNSs can be a promising starting point to tune/engineer silver-based surfaces with foremost applications in the biomedical and industrial sectors.

## Figures and Tables

**Figure 1 ijerph-17-07891-f001:**
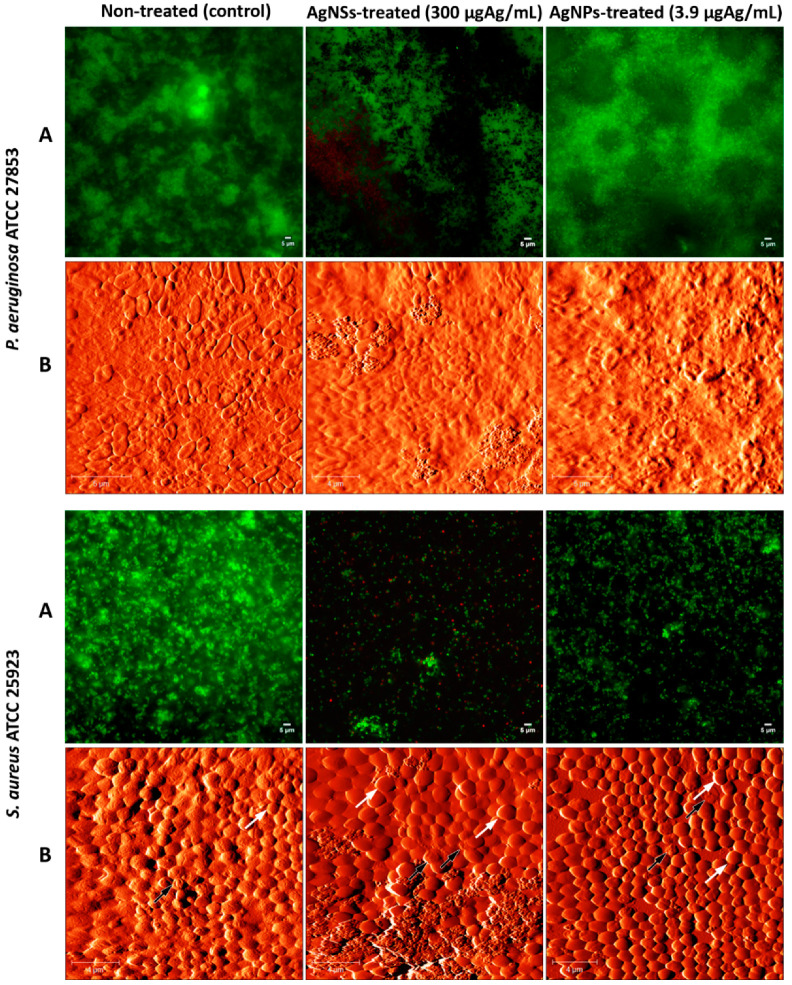
Qualitative evaluation of biofilms treated with AgNSs and AgNPs. Live/Dead images (**A**) and AFM images (**B**) of 24-h biofilms of *P. aeruginosa* ATCC 27853 and *S. aureus* ATCC 25923 non treated (controls), and treated with AgNSs (300 μg_Ag_/mL) or AgNPs (3.9 μg_Ag_/mL) for further 24 h. Some damaged *S. aureus* cells are highlighted by black arrows, and healthy cells by white arrows.

**Figure 2 ijerph-17-07891-f002:**
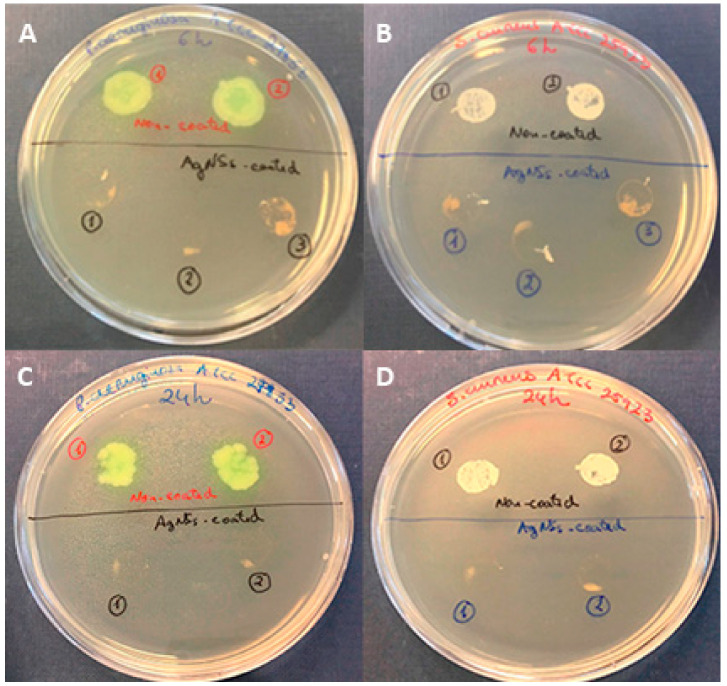
Comparison of the biocidal/killing effects of AgNSs-coated surfaces versus the non-killing effect of AgNSs non-coated surfaces against *P. aeruginosa* ATCC 27853 (**A**,**C**) and against *S. aureus* ATCC 25923 (**B**,**D**) after 6 h (**A**,**B**) or 24 h (**C**,**D**) of contact of the inoculated bacteria with the surface, at 37 °C in the dark.

**Figure 3 ijerph-17-07891-f003:**
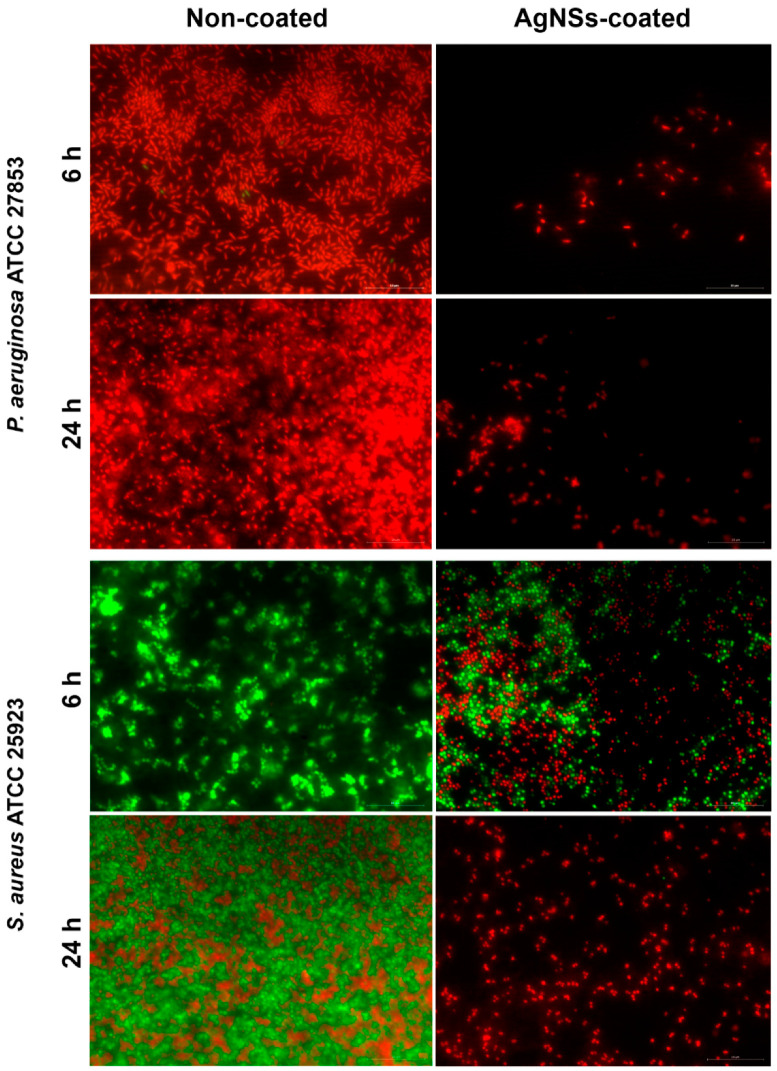
Live/Dead staining images of *P. aeruginosa* ATCC 27853 and of *S. aureus* ATCC 25923 deposited on non-coated and AgNSs-coated surfaces for 6 and 24 h.

**Figure 4 ijerph-17-07891-f004:**
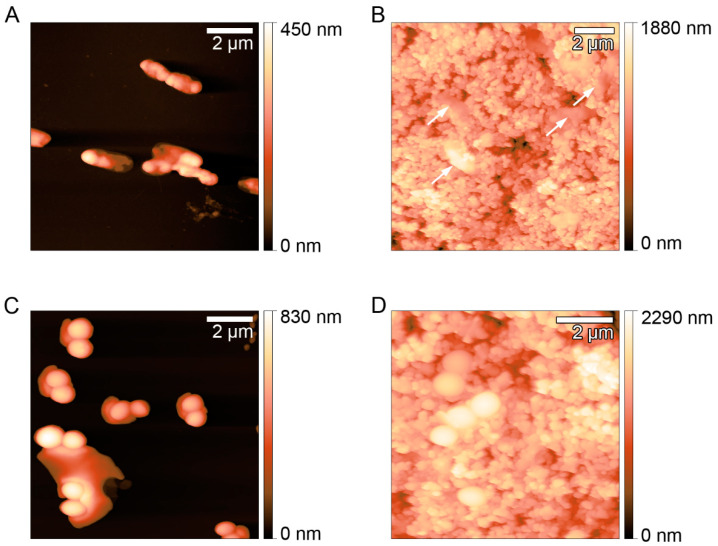
AFM images of *P. aeruginosa* ATCC 27853 (**A**,**B**) and of *S. aureus* ATCC 25923 (**C**,**D**) deposited on non-coated (**A**,**C**) and AgNSs-coated surfaces (**B**,**D**) after 24 h. Arrows point to the *P. aeruginosa* cells, clearly damaged.

**Table 1 ijerph-17-07891-t001:** Minimum inhibitory concentration (MIC) values of AgNSs determined in MHBII and TSB media against Gram-positive and Gram-negative strains.

	MHBII		TSB	
	nmol_AgNSs_·dm^−3^	μg_Ag_/mL	nmol_AgNSs_·dm^−3^	μg_Ag_/mL
*E. coli* ATCC 25922	0.25	600	0.125	300
*P. aeruginosa* ATCC 27853	0.5	1200	0.125	300
*S. aureus* ATCC 25923	0.25	600	0.125	300
*E. faecalis* ATCC 29212	0.125	300	0.125	300

**Table 2 ijerph-17-07891-t002:** MIC values of AgNSs, AgNPs and AgNO_3_ in TSB medium against several strains, including multidrug-resistant isolates of *P. aeruginosa* (PA006 and Pa4) and of *S. aureus* (Sa1 and SA007).

	AgNSs			AgNPs			AgNO_3_	
	nmol_AgNSs_·dm^−3^	μg_Ag_/mL	[Ag] AgNSs/AgNO_3_	nmol_AgNPs_·dm^−3^	μg_Ag_/mL	[Ag] AgNPs/AgNO_3_	nmol_AgNO3_·dm^−3^	μg_Ag_/mL
*E. coli* ATCC 25922	0.125	300	3.33 × 10^4^	0.5	15.5	1.72 × 10^3^	80	0.009
*P. aeruginosa* ATCC 27853	0.125	300	3.33 × 10^4^	>0.5	>15.5	>1.72 × 10^3^	80	0.009
*S. aureus* ATCC 25923	0.125	300	1.76 × 10^4^	>0.5	>15.5	>9.12 × 10^2^	160	0.017
*E. faecalis* ATCC 29212	0.125	300	8.57 × 10^3^	>0.5	>15.5	>4.43 × 10^2^	310	0.035
PA006	0.125	300	3.33 × 10^4^	0.25	7.7	8.56 × 10^2^	80	0.009
Pa4	0.25	600	3.53 × 10^4^	>0.5	>15.5	>9.12 × 10^2^	160	0.017
Sa1	0.25	600	1.71 × 10^4^	>0.5	>15.5	>4.43 × 10^2^	310	0.035
SA007	0.25	600	3.53 × 10^4^	0.5	15.5	9.12 × 10^2^	160	0.017

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
