# Peer review of "Silver Nanostars-Coated Surfaces with Potent Biocidal Properties"

_ijerph, 2020, doi:10.3390/ijerph17217891_

Round 1
Reviewer 1 Report
In this study, the authors investigate the antimicrobial properties of star-shaped silver nanoparticles (AgNSs) against relevant biofilm forming bacterial species. Synthesised AgNSs show lower antibacterial performance than typical spherical silver nanoparticles (AgNPs), per Ag weight unit, against planktonic bacteria in solution, however, they still display an effective bactericidal effect when used as a surface coating.
The manuscript is clear and easy to follow, and the amount experimental work is fair. There is, however, a clear lack of quantitative analysis across the study, which would definitely be beneficial to show the performance of this antimicrobial strategy.
Overall, the experimental evidence is just enough to support the thesis of the authors, but the data interpretation and the discussion require major revisions on the text for this work to be considered for publication.
To be accepted for publication only after these are addressed:
- Table 2 at Line 192:
Typos in nmol/dm3 concentration labels. Same labels are shown for AgNSs, AgNPs and AgNO3 (i.e. nmol AgNSs/dm3).
- Please correct them and state the compound they are associated to ( i.e nmol AgNPs/dm3 and nmol AgNO3/dm3).
- In addition, please add vertical borders to the table to separate the different compounds.This will help the reader to understand which values correspond to each compound.
These corrections are critical since this table is directly involved in the following discussion starting at line 194
- Lines 194-211:
Based on the results at Table 2, authors discuss the MIC values obtained for AgNSs and AgNPs, stating that “MIC values of the AgNSs were 20 to 40 times higher than those of the AgNPs” and point out that these results are “contradictory given the fact that AgNS suspension had a higher concentration of silver than AgNPs suspension”.
If data at Table 2 are correct, MIC molar concentration of AgNSs is always 2-4 times lower than that of AgNPs. Since the concentration of Ag is higher for AgNS than AgNPs a lower MIC is expected for AgNS, as demonstrated by the results in Table 2.
Therefore, these results are not contradictory, as stated by the authors, but expected for these two set of NPs.
- Please modify discussion in lines 194-199 accordingly to correct the points mentioned above.
Despite the revisions mentioned above, the hypotheses (i) and (ii) at lines 201-202 are still valid, considering that the Ag concentration at MIC value is 20-40 times higher for AgNS than AgNPs.
In addition, even if these results might potentially not be explainable considering exclusively the amount of silver ions on the liquid media, such a measurement are not performed by the authors, and the real silver ion release from the AgNS is still an open question, as stated by the authors in (i) line 199.
- Please modify discussion in lines 209-211 accordingly to consider the points mentioned above. Alternatively, if this data is available, please add it here.
- Line 238-239 and Figure S4 in Supplementary Material
Authors estimate a >98% surface coverage for AgNS coating on glass coverslips.
- Could authors provide a quantitative analysis and/or further data i.e. more SEM images that support this estimation?
- In addition to surface coverage, please provide an estimation/observation on the stability of the coating, in terms of surface coverage, after rinsing or after contact with other surfaces?
This information is relevant for the interpretation of antibacterial assays in order to assess the impact of rinsing on the final number of bacteria attached on the coated surfaces.
- Lines 241-244 and Figure 2 in line 246:
Authors state that, using the “touch test”, viable bacteria could not be recovered from coated surfaces after 6h and 24h. However, images in Figure 2 show some “stains” on agar plates as a result of the “touch test” performed for AgNSs coated samples.
- Could authors clarify what those stains are? Are they P. aeruginosa/S. aureus? Are them caused by growth of other environmental bacteria present on the samples? Are they simply caused by the AgNS coatings detached from the samples?
- Lines 250-294, Figures 3 and 4
The antibacterial effect of AgNS coating is evaluated using two complementary techniques, Live/dead staining and AFM.
In the Live/dead assay, for the tested strains, authors observe two distinct effects caused by AgNS antibacterial coatings (i) compromised viability of attached bacteria and (ii) reduction on the total number of attached bacteria.
Even if only a qualitative analysis of the images was performed, the live/dead images show a clearly higher fraction of dead cells for S. aureus. This difference between control and coated samples is not displayed for P. aeruginosa, but the authors provide a sensible explanation as well as evidence from the literature to explain these results.
Regarding the reduction on the total number of attached bacteria on the coated samples relative to the controls, authors claim that, on the control samples, bacteria were able to grow and proliferate. Given the initial high bacterial concentration of the inoculum and the low initial volume (50ul), which could be susceptible to evaporation, as the authors point out, it is unlikely that bacteria can multiply to that extent. The evaporation process and the bacterial growth rate will be very sensitive to the incubation conditions chosen for the selected time periods of 6h and 24h.
- A detailed description of the humidity and temperature conditions chosen for inoculated sample incubation is required. Please add it in the Materials and Methods section.
Alternatively, the reduction in the total number of cells attached to the coated surfaces could be caused by mechanical instability of the coating, that could result in the detachment of AgNSs from the surface, as well as bacteria, during rinsing step in live/dead assay.
- Please comment this based on the coating stability estimation performed as requested for revision 3.
A direct comparison between the antibacterial activity after 24h measured using live/dead assay and AFM shows that in control samples, P. aeruginosa are stained red (“dead”) in fluorescence microscopy but display intact membranes in AFM. On the other hand, fluorescence microscopy shows “dead” S. aureus on coated samples, however, morphological AFM analysis reports a “quite intact morphology” of S. aureus’ cell membrane.
- Please comment why live/dead and AFM seem to contradict each other, even though both assays probe the condition of the bacterial membrane, and state, if any, what technique provides the most reliable results for this kind of test.
Additional minor revision:
- Lines 217-218
Rational behind AgNPs concentration choice is not evident.
- What is the rational of choosing 0.125 nmol/dm3 sub-MIC concentration for AgNPs against resistant bacterial biofilms?
- Figure 1 in line 232-233:
Qualitative interpretation of AFM images is not clear.
- Please add inset arrows in AFM images, as in Figure 4B, to point out healthy and compromised bacteria?
Author Response
Point 1: “1. Table 2 at Line 192:
Typos in nmol/dm3 concentration labels. Same labels are shown for AgNSs, AgNPs and AgNO3 (i.e. nmol AgNSs/dm3).
- Please correct them and state the compound they are associated to (i.e nmol AgNPs/dm3 and nmol AgNO3/dm3).”
Authors: Indeed, labels were typed wrongly. We have corrected them accordingly. We thank the reviewer for alerting us for this correction.
Point 2: “• In addition, please add vertical borders to the table to separate the different compounds.This will help the reader to understand which values correspond to each compound.”
Authors: We have added the vertical borders to the table as suggested.
Point 3: “2. Lines 194-211:
Based on the results at Table 2, authors discuss the MIC values obtained for AgNSs and AgNPs, stating that “MIC values of the AgNSs were 20 to 40 times higher than those of the AgNPs” and point out that these results are “contradictory given the fact that AgNS suspension had a higher concentration of silver than AgNPs suspension”.
If data at Table 2 are correct, MIC molar concentration of AgNSs is always 2-4 times lower than that of AgNPs. Since the concentration of Ag is higher for AgNS than AgNPs a lower MIC is expected for AgNS, as demonstrated by the results in Table 2.
Therefore, these results are not contradictory, as stated by the authors, but expected for these two set of NPs.
- Please modify discussion in lines 194-199 accordingly to correct the points mentioned above.”
Authors: The reviewer is totally right. The data at Table 2 are correct, however, the explanation in the text was quite misleading, in fact. MIC molar concentration of AgNSs is always 2-4 times lower than that of AgNPs. It is the concentration of silver, [Ag], in AgNSs that is 20 to 40 higher in respect to the concentration of Ag in AgNPs. In fact, we thank the reviewer for alerting us for this correction. We have rephrased the sentences (lines 205-210) to achieve accuracy.
Point 4: “Despite the revisions mentioned above, the hypotheses (i) and (ii) at lines 201-202 are still valid, considering that the Ag concentration at MIC value is 20-40 times higher for AgNS than AgNPs.
In addition, even if these results might potentially not be explainable considering exclusively the amount of silver ions on the liquid media, such a measurement are not performed by the authors, and the real silver ion release from the AgNS is still an open question, as stated by the authors in (i) line 199.
- Please modify discussion in lines 209-211 accordingly to consider the points mentioned above. Alternatively, if this data is available, please add it here.”
Authors: We have modified the discussion (lines 218-225) accordingly.
Point 5: “3. Line 238-239 and Figure S4 in Supplementary Material
Authors estimate a >98% surface coverage for AgNS coating on glass coverslips.
- Could authors provide a quantitative analysis and/or further data i.e. more SEM images that support this estimation?”
Authors: We have used the software ImageJ to analyze SEM images. Based on the analysis of images obtained using a lower magnification, such as the new image added to Figure S4 in Supplementary Material, the software estimated that 98.5% (S.D. 0.62 %) was covered by the film of stars. We have inserted that information in the manuscript (lines 260-261) and we have added a new image to Figure S4.
Point 6: “• In addition to surface coverage, please provide an estimation/observation on the stability of the coating, in terms of surface coverage, after rinsing or after contact with other surfaces?”
Authors: The coating was mechanically stable towards rinsing (3-4 times of rinsing did not cause alterations). The surface could, however, be damaged in contact with other surfaces and detached by scraping. This information was added to lines 261-262.
Point 7: “4. Lines 241-244 and Figure 2 in line 246:
Authors state that, using the “touch test”, viable bacteria could not be recovered from coated surfaces after 6h and 24h. However, images in Figure 2 show some “stains” on agar plates as a result of the “touch test” performed for AgNSs coated samples.
- Could authors clarify what those stains are? Are they P. aeruginosa/S. aureus? Are them caused by growth of other environmental bacteria present on the samples? Are they simply caused by the AgNS coatings detached from the samples?”
Authors: Such ‘stains’ are not bacterial growth. They were caused because AgNS coatings detached a little when in contact with the humid agar for 30 seconds. A sentence was inserted to add that information (lines 268-270).
Point 8: “5. Lines 250-294, Figures 3 and 4
The antibacterial effect of AgNS coating is evaluated using two complementary techniques, Live/dead staining and AFM.
In the Live/dead assay, for the tested strains, authors observe two distinct effects caused by AgNS antibacterial coatings (i) compromised viability of attached bacteria and (ii) reduction on the total number of attached bacteria.
Even if only a qualitative analysis of the images was performed, the live/dead images show a clearly higher fraction of dead cells for S. aureus. This difference between control and coated samples is not displayed for P. aeruginosa, but the authors provide a sensible explanation as well as evidence from the literature to explain these results.
Regarding the reduction on the total number of attached bacteria on the coated samples relative to the controls, authors claim that, on the control samples, bacteria were able to grow and proliferate. Given the initial high bacterial concentration of the inoculum and the low initial volume (50ul), which could be susceptible to evaporation, as the authors point out, it is unlikely that bacteria can multiply to that extent. The evaporation process and the bacterial growth rate will be very sensitive to the incubation conditions chosen for the selected time periods of 6h and 24h.
- A detailed description of the humidity and temperature conditions chosen for inoculated sample incubation is required. Please add it in the Materials and Methods section.”
Authors: We agree with the reviewer’s comment. Given the small volume (50 µL) loaded on the surface, and in order to prevent fast evaporation, we have inserted (at time 0h) in the incubator a petri dish full of water in order to humidify the atmosphere and retard evaporation. We did not assess the concrete humidity inside the chamber. The temperature of incubation was 37 ℃. We have noticed that the surfaces, in which the inoculum (50 µL) was loaded, were still very wet after 6 h and were dry after 24 h. Therefore, we had the following information in the materials and methods section, in lines 165-166: “at 37 °C in a humidified incubator”.
Point 9: “Alternatively, the reduction in the total number of cells attached to the coated surfaces could be caused by mechanical instability of the coating, that could result in the detachment of AgNSs from the surface, as well as bacteria, during rinsing step in live/dead assay.
- Please comment this based on the coating stability estimation performed as requested for revision 3.”
Authors: AgNSs coating was stable during rising and to several rinsing cycles. The coating could only be detached by applying a physical force such as scraping. We could assure that the coating was still intact after rinsing. Therefore, attached bacteria were not more washed away in AgNSs-coated surfaces than were from non-coated surfaces.
Point 10: “A direct comparison between the antibacterial activity after 24h measured using live/dead assay and AFM shows that in control samples, P. aeruginosa are stained red (“dead”) in fluorescence microscopy but display intact membranes in AFM. On the other hand, fluorescence microscopy shows “dead” S. aureus on coated samples, however, morphological AFM analysis reports a “quite intact morphology” of S. aureus’ cell membrane.
- Please comment why live/dead and AFM seem to contradict each other, even though both assays probe the condition of the bacterial membrane, and state, if any, what technique provides the most reliable results for this kind of test.”
Authors: In fact, there are apparent contradicting results by both techniques and that we reasonably tried to explain. We were surprised by seeing P. aeruginosa control samples staining red, especially because by AFM, most cells showed intact membranes and also because according to the “touch-test” method they were viable. For that, we had presented a hypothesis in lines 289-293. On the other hand, a possible explanation for the discrepancy between the “dead” S. aureus by fluorescence microscopy and the S. aureus with “quite intact morphology” by AFM was provided in lines 316-321.
All techniques harbor drawbacks, which are more or less marked depending on the type and structural characteristics of the strain being imaged for instance (e.g. Gram-positive vs Gram-negative). Therefore, we sought to use 3 techniques so that we could be sure of the antimicrobial effect of the AgNSs-coated surfaces. Herein, all techniques prove to be useful to reach solid conclusions.
Point 11: “Additional minor revision:
- Lines 217-218
Rational behind AgNPs concentration choice is not evident.
- What is the rational of choosing 0.125 nmol/dm3 sub-MIC concentration for AgNPs against resistant bacterial biofilms?”
Authors: We have chosen that sub-MIC concentration, because we could not determine the exact MIC for P. aeruginosa and S. aureus, MIC >0.5. And we could not increase the range of concentrations because we started from a stock of 1 nmol·dm-3. Given that limitation, and because at the time we did not know the silver concentration in the AgNPs suspension and if it was much alike or different from the silver concentration in AgNSs, we decided to try to use the AgNPs at its sub-MIC, 0.125 nmol·dm-3. More information was added in line 232.
Point 12: “7. Figure 1 in line 232-233:
Qualitative interpretation of AFM images is not clear.
- Please add inset arrows in AFM images, as in Figure 4B, to point out healthy and compromised bacteria?”
Authors: Arrows were added to AFM images of S. aureus biofilms. Accordingly, we provide a new Figure 1. In P. aeruginosa biofilms, bacterial cells were more difficult to be observed within the biofilm hampering the analysis of their traits.

Reviewer 2 Report
Dear authors,
I have read your manuscript entitle: Silver nanostars-coated surfaces with potent biocidal properties. Regarding your manuscript, I have some questions that must be clarified.
1. Page 2 (lines 71-75) - the authors mention that Ag+ can extend antibacterial effects. I cannot disagree with such a statement. Still, when thinking about the applicability of Ag as a biocidal material, Ag+ is always an issue, especially when releasing to the environment. (a) I would encourage the authors to include a discussion on the topic. (b) Also, what is the added value of Ag nanostars when compare with other systems?
2. Page 4 (lines 181-186) - I would need a better hypothesis rather than the physicochemical properties of nanoparticles. The argument is too vague. Ther might be something else, such as stability in suspension over prolonged exposure times.
3. Page 5 (table 2) - there are quite some numbers on Ag concentration. However, there is no comparison to human exposure, aquatic exposure, or even consumption when Ag released. It would be helpful to compare such concentrations to the allowed concentrations of the public health system.
4. A general remark - (a) the authors fail to explain the differences observed for S. aureus and P. aeruginosa when exposed to the Ag nanostars and control. (b) the authors fail to explain the possible mechanism affecting the formed biofilm when exposed to Ag.
5. Page 10 (lines 306-307) - I do not understand the conclusion. Could you elaborate? - Think about How? and the Why? of the statement.
Author Response
Point 1: “1. Page 2 (lines 71-75) - the authors mention that Ag+ can extend antibacterial effects. I cannot disagree with such a statement. Still, when thinking about the applicability of Ag as a biocidal material, Ag+ is always an issue, especially when releasing to the environment. (a) I would encourage the authors to include a discussion on the topic. (b) Also, what is the added value of Ag nanostars when compare with other systems?”
Authors: We have added some information regarding the impact of Ag+ release into the environment (lines 76-84). Regarding question in (b), Ag nanoparticles in general act as "Ag+ reservoirs" (under the form of Ag0, until oxidation and release as Ag+, the bactericidal agent). The star-shaped nanoparticles in specific, present a higher surface area to volume ratio than other shapes. Since the Ag0 oxidation to Ag+ is a surface phenomenon, with the same volume (= mass = amount) of silver we achieve more active nanoparticles.
Point 2: “2. Page 4 (lines 181-186) - I would need a better hypothesis rather than the physicochemical properties of nanoparticles. The argument is too vague. Ther might be something else, such as stability in suspension over prolonged exposure times.”
Authors: AgNSs suspension tends to deposit after a while in the bottom of the tube, however, it quickly resuspends after vortexing for 3-5 seconds. When the suspension was used to prepare dilutions in both media, it was redispersed equally in each medium. Nonetheless, after a while more than 1 h, we could see that it deposited in the bottom of the wells when either TSB or MHBII was used. Despite that has occurred in both media, we found differences in the MIC determined depending on the medium used. Therefore, those differences are most likely to be due to the media composition and to the interaction of certain components (mostly proteins) with the nanoparticles, altering their physicochemical properties. That would explain the different MICs values obtained in the two media.
Point 3: “3. Page 5 (table 2) - there are quite some numbers on Ag concentration. However, there is no comparison to human exposure, aquatic exposure, or even consumption when Ag released. It would be helpful to compare such concentrations to the allowed concentrations of the public health system.”
Authors: Indeed, silver itself is classified as an environmental hazard because it is toxic, persistent and can bioaccumulate under some circumstances. However, silver is not especially toxic to humans or other mammals. Therefore, the environmental risk of silver must and has been investigated.
In this context, we quote relevant information from Samuel N. Luoma, (Silver nanotechnologies and the environment: old problems or new challenges, Project on Emerging Nanotechnologies of the Woodrow Wilson International Center for Scholars, 2008 - 66 pages). “However, the environmental risks from silver itself might be mitigated by a propensity of the silver ion to form strong complexes that are apparently of very low bioavailability and toxicity. In particular, complexes with sulfides strongly reduce bioavailability under some circumstances. It is not yet clear to what extent such speciation reactions will affect the toxicity of nanosilver. If organic/sulfide coatings, or complexation, in natural waters similarly reduce bioavailability of nanosilver particles, the risks to natural waters will be reduced. Nevertheless, the environmental fate of nanosilver will depend upon the nature of the nanoparticle. Nanoparticles that aggregate and/or associate with dissolved or particulate materials in nature will likely end up deposited in sediments or soils. The bioavailability of these materials will be determined by their uptake when ingested by organisms.
Toxicity is determined by the internally accumulated, bioavailable nanosilver in each organism, not just the total concentration in the environment. However, biological factors also influence toxicity. If the organism can sequester the silver in forms that are not toxic (detoxification), then all the internal silver will not be biologically active and the contaminant will be less toxic. The forms of nanosilver that will be most toxic are those that are taken up readily from the environment, excreted slowly and/or are not sequestered internally in a nontoxic form.”
Even though the Ag concentration in the AgNSs at the MIC is 300 µg/mL, we do not know how much of the silver present in AgNSs is being released, but surely, less than that. However, such a measurement was not performed and, thus, the real silver ion release from the AgNS is still an open question (as stated in lines 222-223).
A sentence and a reference were added in the manuscript (lines 245-247).
Point 4: “4. A general remark - (a) the authors fail to explain the differences observed for S. aureus and P. aeruginosa when exposed to the Ag nanostars and control. (b) the authors fail to explain the possible mechanism affecting the formed biofilm when exposed to Ag.”
Authors: Regarding (a), no significant differences were observed between AgNSs-treated biofilms and the respective non-treated biofilms (controls), for both strains assayed, S. aureus and P. aeruginosa. A new sentence was inserted in the manuscript (lines 238-240). Regarding (b), since no marked effect was observed on AgNSs-treated biofilms, only a slight increase in the number of dead cells within the treated biofilms, no specific mechanism of AgNSs on the biofilm can be hypothesized. The effect of AgNSs is on the bacterial cell, causing dead, and not in affecting the overall biofilm.
Point 5: “5. Page 10 (lines 306-307) - I do not understand the conclusion. Could you elaborate? - Think about How? and the Why? of the statement.”
Authors: We have rephrased the sentence accordingly (lines 333-335).

Round 2
Reviewer 2 Report
Thank you to the authors. I am pleased with most of the answers.